# Selective Removal of As(V) Ions from Acid Mine Drainage Using Polymer Inclusion Membranes

**Iwona Zawierucha [1],\*** , **Anna Nowik-Zajac [1]** and **Grzegorz Malina [2]**

[1] Institute of Chemistry, Jan Dlugosz University in Czestochowa, 42-200 Czestochowa, Poland; a.zajac@ajd.czest.pl

[2] Department of Hydrogeology and Engineering Geology, AGH University of Science and Technology, Mickiewicza 30, 30-059 Cracow, Poland; gmalina@agh.edu.pl

\* Correspondence: i.zawierucha@ajd.czest.pl; Tel.: +48-343614918-(251)

**Abstract:** Acid mine drainage (AMD) is globally recognized as one of the environmental pollutants of the priority concern due to high concentrations of toxic metals and sulfates. More rigorous environmental legislation requires exploitation of effective technologies to remove toxic metals from contaminated streams. In view of high selectivity, effectiveness, durability, and low energy demands, the separation of toxic metal ions using immobilized membranes with admixed extractants could ameliorate water quality. Cellulose triacetate based polymer inclusion membranes (PIMs), with extractant and plasticizer, were studied for their ability to transport of As(V) ions from synthetic aqueous leachates. The effects of the type and concentration of extractant, plasticizer content, and sulfuric acid concentration in source phase on the arsenic removal efficiency have been assessed. Under the best of applied conditions, PIM with Cyanex 921 as extractant and *o*-nitrophenyl octyl ether (*o*-NPOE) as plasticizer showed high repeatability and excellent transport activity for selective removal of As(V) from AMD.

**Keywords:** arsenic; acid mine drainage; polymer inclusion membranes

## 1. Introduction

Mining of minerals is associated with acid drainage problems that can give rise to long-term impairment to aquatic environment and biodiversity. Acid mine drainage (AMD) is produced when sulfide-bearing material is comes into contact with oxygen and water [1]. It can be characterized as acidic water (pH < 2) with high concentrations of sulfate and various metals and metalloids [2–4]. Penetration of AMD into surface and ground waters is a critical environmental issue and often results in biotic degradation of water bodies around the abandoned mines through, e.g., direct toxicity, changes of habitats, and in the nutrient cycle, leading to unsuitability of water resources for use in housing, agriculture, and industry [5–7]. The acidification and leaching of toxic elements like metals and arsenic are mainly reasons for the environmental degradation. The region of AMD sites includes polluted soil and groundwater that threatens the flora and fauna [8]. Acidic (ground)water can be detrimental to both aquatic and terrestrial organisms [9], and it may trigger the mobilization of diverse metals and metalloids [10]. As a result, the AMD pose an environmental and health threats and a major challenge to clean up hazardous pollutants [11].

Arsenic (As) being a common AMD pollutant is a naturally occurring metalloid very mobile in the environment. Its mobility largely depends on the parent mineral form, oxidation state, and mobilization mechanisms [12]. Arsenic exists in the −3, 0, +3 and +5 oxidation states; however, arsenite (As(III)) and arsenate (As(V)) are the most frequently found in water [13–16]. Due to slow redox transformations, they are present in both reduced and oxidized environments. In the case of anaerobic (reduced)

conditions, e.g., groundwater and sediment, arsenic is present mainly as arsenite, while arsenate is dominant in oxidative environments (e.g., surface waters) [17]. The pH also plays an important role in determining the state of arsenic in aqueous solutions [18]. Arsenic is well known for its toxic effects [8,19,20]. It may lead to the noxious impacts on human health like skin defects, disturbances of blood circulation, neurological problems, diabetes, difficulties with breathing, and hepatic and renal abnormalities [21].

Typical processes for treating AMD rely on arsenic removal via oxidation, precipitation, and separation [22,23]. Techniques applied conventionally to remove arsenic species from water include, oxidation, coagulation–flocculation, ion exchange adsorption, and membrane techniques [24]. Recently, advances in nanoscience and nanotechnology have triggered the development of various nanomaterials as novel adsorbents of heavy metals and arsenic from aqueous solutions, including AMD [25,26]. Bioremediation of arsenic contaminated soil and (ground)water is advantageous related to its environmental affinity and possible cost benefits. It is based on microbial activity to remove, mobilize, and hold arsenic through sorption, biomethylation–demethylation, complexation, coprecipitation, and oxidation–reduction processes [27].

The current development of AMD passive treatment systems has recently been reviewed by Skousen et al. (2017) [7]. Long-term, effective passive AMD treatment systems based on geogenic (geochemical and biological) processes allow for acidity neutralization, oxidation, or metals and arsenic reduction and precipitation, and are suitable for small and medium AMD discharges. Constructed wetlands, including vertical flow wetlands and bioreactors as examples of passive biological treatment technologies, are generally related to microbial activity and can use organic matter to stimulate microbial sulfate reduction and contaminants adsorption. In the case of passive geochemical systems, on the other hand, alkalinity-generating materials, e.g., limestone, come into contact with AMD or with uncontaminated water above the AMD discharge [7].

Sulfate reducing bacteria (SRB) are used for in situ AMD treatment; however, the choice of organic substrate as a carbon source affects the process efficiency [28]. Permeable reactive barriers (PRB) are considered innovative to treat AMD [29]. To treat AMD with metals contamination the PRB is generally composed of solid organic matter, such as, municipal/leaf compost and wood chips/sawdust. Organic matter enhances the proliferation of sulfate-reducing bacteria that may reduce sulfate to sulfide, and the subsequent formation of insoluble metal sulfides [30].

The effective removal of metal ions from aqueous waste streams can be obtained by applying a polymer inclusion membranes (PIMs) technique. Separation of metal ions using this membrane system is a multi-stage process including extraction of the substance from the source phase, diffusion of the resulting complex through the membrane and re-extraction into the receiving phase [31]. PIM is created in the form of thin, flexible, and stable film by molding a solution containing an extractant (an ion carrier), a plasticizer and a base polymer, such as cellulose triacetate (CTA) or polyvinyl chloride (PVC). The obtained membrane provides the barrier dividing source and receiving aqueous phases [31–35].

The PIM separation properties are advantageous due to high membrane selectivity with its increased durability. The membrane stability is due to the immobilization of extractant in the solid polymer matrix, while its high permeability is provided by addition of the plasticizer [31]. In addition, PIMs are characterized by mechanical strength comparable to that of filtration membranes. An important feature of PIM-based system is the possibility of modifying membrane composition to gain the appropriate selectivity and separation efficiency. The other advantages of PIM technique include ease of operation and minimum use of hazardous chemicals [36,37].

Guell et al. (2011) [38] studied the transport of arsenate from water samples across the PIM composed of CTA and Aliquat 336, and using 0.1 M NaCl as a receiving phase. They found that As(V) transport efficiency was not affected by presence of co-existing anions in natural waters. Vera et al. (2018) [39] using PIM made of 52% CTA, 48% Aliquat 336 (*w/w*), and 2 M NaCl as a receiving phase obtained As(V) removal efficiency within the range of 53–81% in different water samples. Moreover, they noted that transport was not dependent on the kind of polymer and the membrane thickness [39]. The transport of arsenate ions has been also investigated by De Lourdes Ballinas et al. (2004) [40] using a PIM based on CTA and dibutyl

butyl phosphonate (DBBP) as the carrier. High arsenic recovery factor (90% in 800 min) was obtained for an initial As(V) concentration of 3000 mg/L and the receiving solution contained 2 M LiCl.

In this work the removal of As(V) ions from synthetic aqueous leachates (sulfuric acid solutions) by transport through PIM has been studied. The membrane consisted of cellulose triacetate (CTA) as the polymeric support, *o*-nitrophenyl octyl ether (*o*-NPOE) as the plasticizer, and solvating extractants as ion carriers. The effects of various parameters on the removal efficiency of arsenic were studied, including the plasticizer content, type, and concentration of a carrier in the membrane, and the sulfuric acid concentration in the source phase. Moreover, in order to extend the applicability of the separation technique, we have evaluated reusability of PIM, and tested the PIM-system for selective removal of arsenic from a real AMD sample.

## 2. Materials and Methods

### 2.1. Reagents

Inorganic chemicals, i.e., arsenate(V) sodium ($Na_2HAsO_4 \cdot 7H_2O$), sodium chloride (NaCl), and sulfuric acid ($H_2SO_4$) of analytical grade were purchased from POCh (Gliwice, Poland). Organic reagents, i.e., cellulose triacetate (CTA), *o*-nitrophenyl octyl ether (*o*-NPOE), and dichloromethane (DCM) were also of analytical grade, purchased from Fluka (Buchs, Switzerland) and used without further purification. Aqueous solutions were prepared with double distillation water, with a conductivity of 0.1 μS/cm. The carriers: DBBP ($CH_3(CH_2)_3P(O)[O(CH_2)_3CH_3]_2$) and Cyanex 921 ($[CH_3(CH_2)_7]_3PO$) were purchased from Sigma-Aldrich (St. Louis, MO, USA).

### 2.2. Preparation Of Polymer Inclusion Membranes and Stability Test

The preparation pf polymer inclusion membranes was carried out based on the procedure previously described [31]. To prepare PIMs, the following solutions were used (in DCM as an organic solvent): cellulose triacetate (CTA) as a support, *o*-nitrophenyl octyl ether as a plasticizer, and DBBP and Cyanex 921 as ionic carriers. The defined volumes of the CTA solution, plasticizer, and carrier were mixed, and this mixture was poured in to a 5.0 cm glass ring attached to a plate glass with CTA—dichloromethane glue. The glass ring was left to rest overnight to DCM evaporate in room temperature. The obtained membrane was disconnected from the glass plate by wetting in cold water. The effective surface area of the membrane was of 4.9 cm$^2$. The membrane thickness was measured with an accuracy of 1.0 μm standard deviation over four readings by A2002M type digital ultrameter from Inco-Veritas (Warsaw, Poland). The average CTA membrane thickness was of 24 μm.

The stability of PIMs was investigated in terms of mass loss, which is related to the leaching of carrier and/or plasticizer. For that, PIMs were immersed in 100 mL of ultrapure water and was shaken for 24 h. Membranes were weighed before and after this procedure, and the mass loss was calculated. The membrane masses were also examined before and after arsenic ions transport.

### 2.3. Transport Studies

Transport experiments were carried out in a permeation cell described in our earlier paper [31] using a two-compartment cell where the membrane film was tightly clamped between the source and receiving phases. The source phase was of an aqueous solution of As(V) in sulfuric acid media (50 cm$^3$) whereas the aqueous receiving phase was 1.0 M sodium chloride (50 cm$^3$). The experiments were performed at a room temperature (23–25 °C) and both source and receiving aqueous phases were agitated at 600 rpm with synchronous stirrers. Samples of aqueous phases were removed periodically via a sampling port with a syringe and analyzed to determine the As(V) concentration. The acidity of both aqueous phases was controlled by pH-meter (multifunctional pH-meter, CX-731 Elmetron, with a combine pH electrode, ERH-136, Hydromet, Poland) [31,35].

Kinetics of the transport process through PIM is described by a first-order reaction in respect to a metal ion concentration [41]:

$$\ln\left(\frac{c}{c_0}\right) = -kt \tag{1}$$

where $c$ is the metal ion concentration (mol/dm$^3$) in the source phase at a given time, $c_0$ is the initial As(V) concentration in the source phase (mol/dm$^3$), $k$ is the rate constant (s$^{-1}$), and $t$ is the transport time (s).

To calculate the $k$ value, a graph of $\ln(c/c_0)$ versus time was plotted. The relationship of $\ln(c/c_0)$ versus time was linear, which was confirmed by high values of determination coefficients (r$^2$), i.e., $\geq 0.99$. (Figure 1). The permeability coefficient *(P)* was calculated as follows [31,35,42]:

$$P = -\frac{V}{A}k \tag{2}$$

where $V$ is the volume of aqueous source phase (m$^3$), and $A$ is the surface area of membrane (m$^2$). The initial flux ($J_0$) in (μmol/m$^2$s) was determined as equal to

$$J_0 = Pc_0 \tag{3}$$

To describe the efficiency of metal ion removal from the source phase, the recovery factor (*RF*) was calculated as

$$RF = \frac{c_0 - c}{c_0} \cdot 100\% \tag{4}$$

The metal ions concentrations were measured by flame atomic absorption spectrometry (Solar 939, Unicam, Munich, Germany) and the sulfate concentration was determined by ion chromatography (861 Advanced Compact IC, Metrohm, Herrisau, Switzerland). The reported values correspond to the average of duplicates, and the standard deviation observed was within 2%.

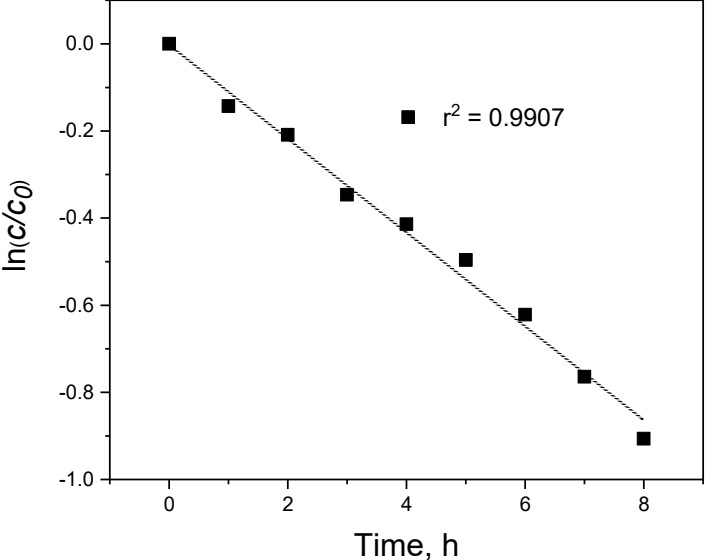

**Figure 1.** The relationship of $\ln(c/c_0)$ vs. time for arsenic transport across polymer inclusion membranes (PIM). Source phase: $1.0 \cdot 10^{-2}$ M As(V), 1.0 M H$_2$SO$_4$; membrane: 19 wt.% of cellulose triacetate (CTA), 3 wt.% of Cyanex 921, 78 wt.% of *o*-NPOE; receiving phase: 1.0 M NaCl.

## 3. Results and Discussion

### 3.1. Kinetics of As(V) Ions Transport through Polymer Inclusion Membranes

The changes of concentration of As(V) ions in the source, receiving and the membrane phases were analyzed to determine their transport through PIM containing Cyanex 921 by creating a concentration profile of the metal as a function of time (Figure 2). The obtained exponential curves of $c/c_0$ to time corroborates the kinetic model of metal ions transport suggested by Danesi et al. (1984) [41]. Figure 2 indicates that the transport of As(V) ions across membrane was speedy and parallel. The lack of the metal ions aggregation in the membrane phase was also noted.

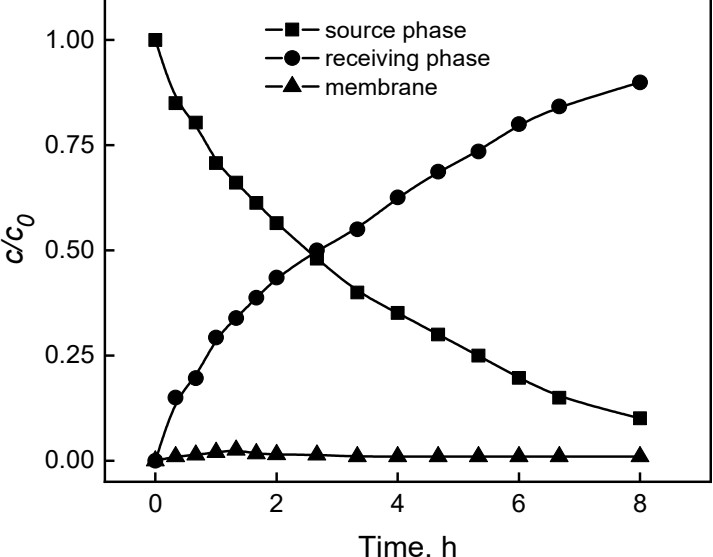

**Figure 2.** The profile of As(V) concentrations in the source phase, membrane and receiving phase during transport through PIM. Source phase: $1.0 \cdot 10^{-2}$ M As(V), 1.0 M $H_2SO_4$; membrane: 19 wt.% of CTA, 3 wt.% of Cyanex 921, 78 wt.% of $o$-NPOE; receiving phase: 1.0 M NaCl.

### 3.2. Effect of Membrane Composition

The transport through PIMs is mainly affected by the following factors: the composition of the membrane and selection of appropriate receiving phase [42]. The effect of the type and content of ion carriers in PIM on As(V) ions transport was studied. Membranes with a predetermined content of CTA (25 mg-19 wt.%) and plasticizer (77 wt.% of $o$-NPOE) were synthesized, while the concentration of carrier (DBBP or Cyanex 921) in the membrane was ranged from 1 to 8 wt.% (from 0.01 to 0.30 M with respect to the plasticizer volume). The $o$-NPPE was chosen as the plasticizer due to its low viscosity and a high dielectric constant. Plasticizer is a solvent for an ion carrier (extractant) and enhances the flexibility and permeability of the membrane. In turn the mechanical strength of the membrane is given by the CTA polymer [43]. The presence of carrier is essential for the facilitated transport of arsenic ions through the membrane. This fact was confirmed by results of blank experiments (in the absence of carrier), which showed no significant flux across PIM containing only the support and plasticizer.

Figure 3 shows that the transport of As(V) increased as the carrier concentration increased. However, this effect stopped at a concentration of 0.15 M (4 wt.%). The fluxes tended to a constant value at higher carriers concentrations, which was probably due to saturation of the membrane pores with metal complex species. These results indicate that the arsenic removal in the PIM system occurred in the following steps. Firstly, the metal ions moved from the source phase to the boundary layer of the membrane. Then, the complex As(V)-extractant was formed at the source solution–PIM interface. Finally, the As(V)-carrier complex transferred through PIM and As(V) was released at the PIM–receiving solution interface [44]. The arsenic transport process occurred in the presence of the carrier (extractant)

contained within the membrane phase, so the coupled facilitated transport took place [45]. The carrier reacted with arsenic ions producing a complex which was transported through the membrane. The pH gradient between source and receiving phases was the driving force for the transport of metal ions across PIM [46]. The highest flux values were of 0.96 for DBBP and 1.30 $\mu mol/m^2 s$ for Cyanex 921. The highest RF values (i.e., 89.0% and 95.8%) were obtained for membranes with the 0.15 M carrier concentration (DBBP and Cyanex 921, respectively) (Figure 4). The lower RF values were for carrier concentrations greater than 0.15 M, which can be explained by the effect of increasing membrane viscosity; a higher viscosity of the membrane limited the diffusion of the As(V)−carrier complex within the membrane phase. Based on these results, for further experiments, the Cyanex 921 as ion carrier was used.

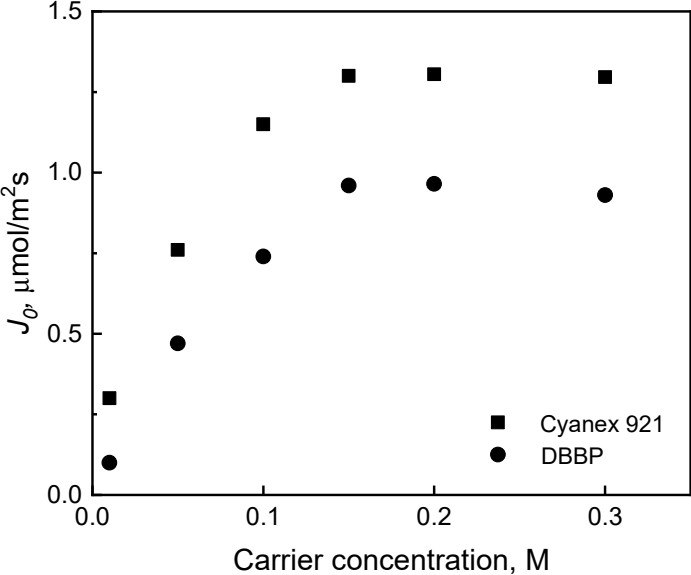

**Figure 3.** The As(V) fluxes vs. ions carriers concentration in the PIM. Source phase: $1.0 \cdot 10^{-2}$ M As(V), 1.0 M $H_2SO_4$; membrane: 19 wt.% of CTA, 77 wt.% of *o*-NPOE; receiving phase: 1.0 M NaCl.

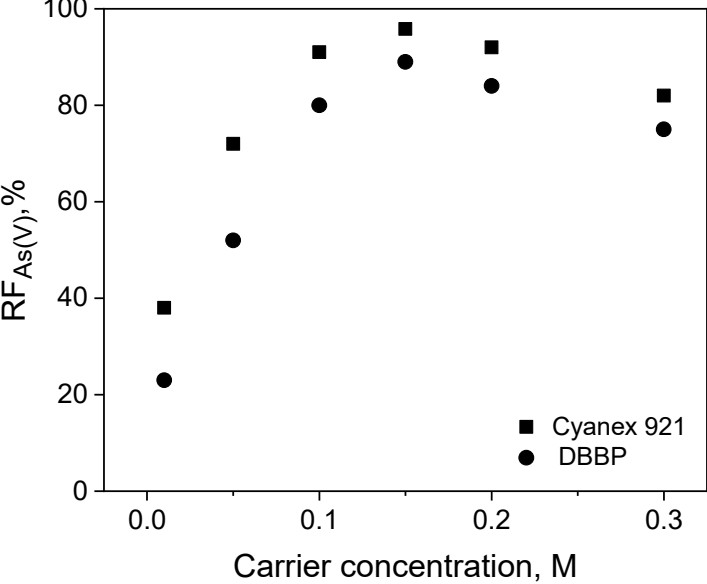

**Figure 4.** Recovery factor (RF) of As(V) ions vs. carrier concentration. Source phase: $1.0 \cdot 10^{-2}$ M As(V), 1.0 M $H_2SO_4$; membrane: 19 wt.% of CTA, 77 wt.% of *o*-NPOE; receiving phase: 1.0 M NaCl.

The Cyanex 921 extractant is composed of tri-n-octylphosphine oxide with a molecular weight of 386 [45]. In a liquid membrane system, Cyanex 921 can extract both acids and metal complexes [34,47]. The decrease in pH of the receiving phase over time indicated that protons $H^+$ from sulfuric acid were transferring to NaCl solution, acidifying it. However, this coextraction did not affect the As(V) stripping.

The effect of the plasticizer amount in PIM on As(V) ions transport was also examined. For this purpose, the membranes with constant carrier concentration of 0.15 M and different contents (0.5–2.0 cm$^3$) of o-NPOE/1.0 g CTA (63–87 wt.%) were prepared. Figure 5 shows the relationship between fluxes and amounts of o-NPOE. With the increasing amount of plasticizer, the fluxes increased. This trend had been kept until reaching a specific value of plasticizer in the membrane (1.0 cm$^3$ o-NPOE/1.0 g CTA, 77 wt.%). A further increase of the plasticizer amount was related to an increase of the thickness of PIM which probably reduced the permeability of the membrane. This resulted in a limiting the diffusion rate of metal complexes through membrane and a visible decrease in transport efficiency.

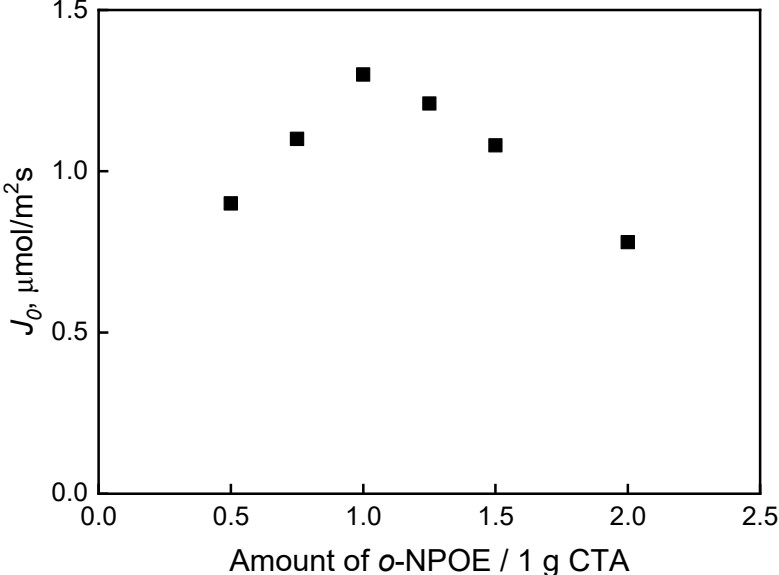

**Figure 5.** The effect of o-NPOE plasticizer on As(V) transport through PIM containing Cyanex 921. Source phase: 1.0·10$^{-2}$ M As(V), 1.0 M H$_2$SO$_4$; membrane: 19 wt.% of CTA, 4 wt.% of Cyanex 921; receiving phase: 1.0 M NaCl.

The highest RF factor of As(V) (95.8%) was achieved by transport through the PIM composed as follows: 19 wt.% of cellulose triacetate as the support, 4 wt.% of Cyanex 921, and 77 wt.% of o-nitrophenyl octyl ether as the plasticizer. Therefore, such PIM composition has been referred to as "optimal" and used in our further tests.

The image (3-D scan) obtained by atomic force microscopy (AFM) for membrane with an "optimal" composition is shown in Figure 6b. The colour intensity represents a vertical profile of the sample; the light areas indicate the highest spikes while dark areas the pores (organic inclusion in CTA support). The pores are noticeable as small well-defined dark regions. On the other hand, the membrane from CTA itself (Figure 6a) is non-porous and only slightly creased. The middle sized pores estimated for the optimal PIM were equal to 0.05 μm.

The stability of "optimal" PIM was investigated in terms of the extent of leaching of its components (the carrier and plasticizer) from the membrane to ultrapure water. Membrane mass loss was found to be 7 ± 1% (n = 2), which indicated good resistance of membrane components to migration from the base polymer. Plasticizer (o-nitrophenyl octyl ether), as a member of phthalate esters group, hydrolyzes below pH 5 and above pH 7. The degree of hydrolysis increases with increasing or decreasing pH and with longer contact times [48]. In our study (at acidic source phase), membrane mass loss after 8 h of As(V) transport was of 9 ± 0.4% (n = 2), which demonstrated that the plasticizer was properly bound

to the polymer. Otherwise, in the case of significant leaching, due to toxicity of the phthalates [49], it is not necessary to use the plasticizer at the pilot or industrial scales.

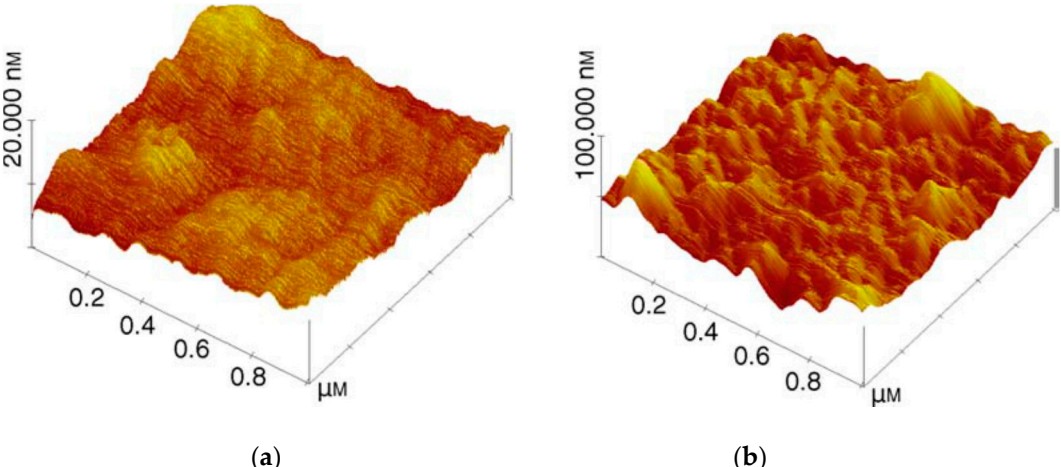

(**a**)                    (**b**)

**Figure 6.** The 3-D image of atomic force microscopy (AFM) for CTA membrane (**a**) and "optimal" polymer inclusion membrane (**b**).

### 3.3. Modification of the Source Phase Acidity

The effect of sulfuric acid concentration in the source phase within the range of 0.01–2.0 M on As(V) ions transport through PIM was examined. As is seen in Figure 7 the arsenic removal efficiency increased with rising acidity of the source phase, reaching a maximum of 95.8% at the $H_2SO_4$ concentration of 1.0 M. This can be explained by the fact that in acidic conditions (pH < 2), As(V) is found in the form of $H_3AsO_4$—a neutral particle that can be removed by Cyanex 921 (solvating extractant) [50]. At low concentrations of $H_2SO_4$, arsenic acid is deprotonated, and thus the removal percentage is lower [47]. High efficiency of arsenic removal using PIM at 1.0 M concentration of $H_2SO_4$ in the source phase was also because the source and receiving phases differed significantly in the pH value. The pH gradient stimulated diffusion of the As(V)−carrier complex through the membrane and, consequently, the efficiency of arsenic separation by the PIM.

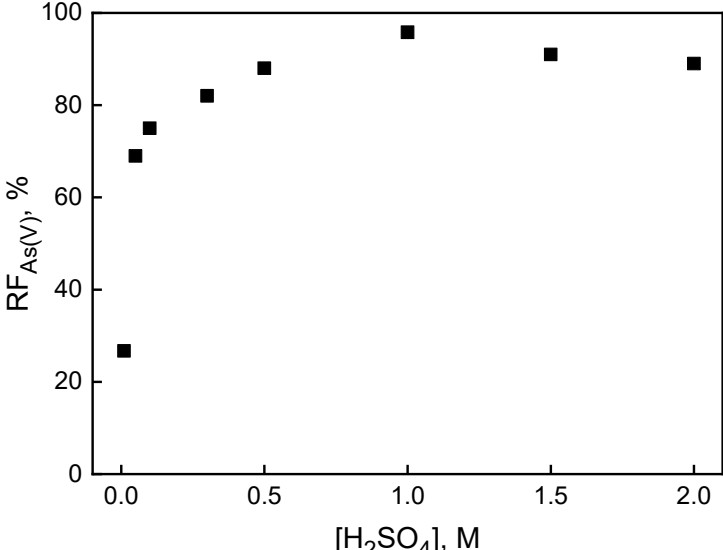

**Figure 7.** Recovery factor (RF) of As(V) ions vs. $H_2SO_4$ concentration in the source phase. Source phase: $1.0 \cdot 10^{-2}$ M As(V); membrane: 19 wt.% of CTA, 4 wt.% of Cyanex 921, 77 wt.% of *o*-NPOE; receiving phase: 1.0 M NaCl.

### 3.4. Membrane Reusability

The reusability, among different properties of PIMs, provides the greatest advantage, ensuring its industrial applicability [35]. In the repeated As(V) transport (eight cycles), the same PIM membrane was used while both aqueous phases were replaced for each cycle. The results indicate (Figure 8) that the efficiency of the PIM is repeatable. The RF values were above 95% in the first four cycles of the PIM (each cycle—8 h), while then slightly decreased (the RF values were ca. 92%). The optimal PIM seems to be effective for multiple use for arsenic separation process.

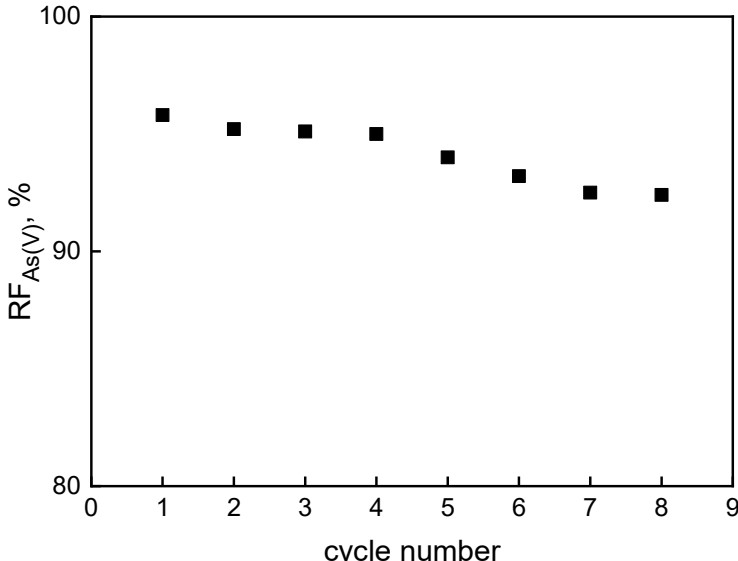

**Figure 8.** Recovery factor (RF) of As(V) vs. cycle number. Source phase: $1.0 \cdot 10^{-2}$ M As(V), 1.0 M $H_2SO_4$; membrane: 19 wt.% of CTA, 4 wt.% of Cyanex 921, 77 wt.% of *o*-NPOE; receiving phase: 1.0 M NaCl.

### 3.5. Selective Removal of As(V) from AMD

The optimal PIM was tested for its applicability for the removal of arsenic from an AMD sample originated from the Wisniowka mining area (south-central Poland) [51]. The AMD was acidic (pH 1.8) with As(V) and sulfate concentrations of 782 and 93,900 mg/dm$^3$, respectively. Concentrations of other metals like: $Fe_{total}$, Mn, Cu, Ni, and Cr were of 34,200, 50, 46, 27, and 20 mg/dm$^3$, respectively. The As(V) transport studies were carried out using the optimal PIM and 1.0 M sodium chloride as the receiving phase. After 8 h of continuous operating time the removal efficiency of arsenic was of 90% (Figure 9). Moreover, the RF values found for sulfate and other metals were below 5%, which prove the selectivity of the system for arsenic.

Microbial treatment based on the use of SRB could be considered as an approach for subsequent remediation of AMD to reducing sulfate, increase pH, and precipitate other metals as metal sulfides. For remediation to occur, SRB require, besides the presence of sulfate as electron acceptors, anaerobic conditions and an outer carbon source as electron donors, typically supplied from a simple organic compound. The process gives hydrogen sulfide and bicarbonate that treat the AMD by creating insoluble metal sulfides [52]. The produced metal sulfide sludge is more dense, faster settling, and less subject to dissolution. In addition, the process induces enough alkalinity to increase the pH without the necessity of any addition of alkaline agents [53].

In addition to microbial treatment, number of methods developed to treat metals contaminated water streams can potentially be applied as post-treatment after removing As from AMD using PIMs and prior to its discharge to the environment or further use. They include: chemical precipitation [54], ion exchange [55], adsorption [56], membrane filtration [57], reverse osmosis (RO) [58], solvent extraction [59], and electrochemical treatment [60]. For example, hydroxides or sulfur-containing precipitation agents commonly used for effective removal of metals from water and wastewater comprise [61]: potassium/sodium thiocarbonate (STC) and

2,4,6-trimercaptotiazine (TMT), and dithiocarbamate (DTC) compounds, e.g., sodium dimethyldithiocarbamate (SDTC), and ligands for permanent metal binding, e.g., 1,3-benzenediamidoethanethiol (BDETH2), 2,6-pyridinediamidoethanethiol (PyDET), a pyridine-based thiol ligand (DTPY), or ligands with open chains containing many sulfur atoms, using of a tetrahedral bonding arrangement around a central metal atom. High removal efficiencies of Cu and Cd ions of 98 and 99%, respectively, using RO were reported by Qdais and Moussa (2004) [62], whereas nanofiltration was capable of removing more than 90% of Cu ions [62]. Comparative sorption studies on the potential of indigenously synthesized carbon nanomaterials for removal of diverse metals from contaminated water streams showed higher sorption of Cd, Pb, Ni, and Zn on nanoporous carbon compared to nanocarbon [63]. The toxic metals like Hg, Pb, Cd, and Cu were also effectively removed from water using $Fe_3O_4$ magnetic nanoparticles coated with humic acid [64].

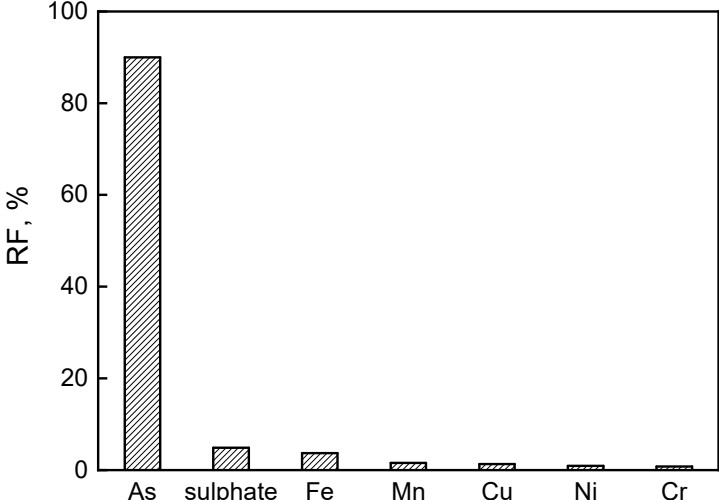

**Figure 9.** The RF values obtained in the competitive PIM transport of arsenic from the AMD sample. Membrane: 19 wt.% of CTA, 4 wt.% of Cyanex 921, 77 wt.% of *o*-NPOE; receiving phase: 1.0 M NaCl.

## 4. Conclusions

The study of As(V) ions transport through polymer inclusion membranes (PIMs) containing solvating extractant as the carrier allowed to specify the factors determining the removal efficiency. The membrane saturation obtained with 4 wt.% of Cyanex 921 for transport of As(V) through PIM gives the highest efficiency (95.8%) when the concentration of $H_2SO_4$ in the source phase was equal to 1.0 M, and 1.0 M sodium chloride was used as the stripping agent. The *o*-nitrophenyl octyl ether proved to be the suitable plasticizer in PIM for the effective arsenic removal, reaching the maximum recovery factor (RF) value for membranes containing 77 wt.% of *o*-NPOE. The optimal PIM composition was found to be 19 wt.% of cellulose triacetate as the support, 4 wt.% of Cyanex 921 as the ion carrier, and 77 wt.% of *o*-nitrophenyl octyl ether as the plasticizer. The transport efficiency of the PIM can be reproduced; thus, the membrane could be multiple used for arsenic separation. Moreover, the obtained results show the good performance of the PIM respected to the selective removal of arsenic from AMD with a separation efficiency of 90%.

**Author Contributions:** Conceptualization, I.Z. and A.N.-Z.; investigation, I.Z. and A.N.-Z.; supervision, G.M.; writing—original draft, I.Z.; writing—review and editing, A.N.-Z. All authors have read and agreed to the published version of the manuscript.

**Funding:** This research was funded by the National Science Centre, Poland (Grant number UMO-2015/19/D/ST10/03214).

**Acknowledgments:** The authors gratefully acknowledge Jan Dlugosz University in Czestochowa for the support provided for the publication of this work.

**Conflicts of Interest:** The authors declare no conflict of interest.

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
