# Peer review of "Selective Removal of As(V) Ions from Acid Mine Drainage Using Polymer Inclusion Membranes"

_minerals, doi:10.3390/min10100909_

Round 1

Reviewer 1 Report

Comments for the Authors

The Authors in Abstract stated that “Under optimal conditions, …”. What kind of optimization method the Authors applied? If none, it means that conditions were the best of applied, not optimal.
The Authors applied defined (various) volumes of CTA, plasticizer and carriers. It means that the total volume was different. What was the time of evaporation because of various volumes of solution, the same or different ?
In the Reagent section, the Authors stated that o-NPOE was applied as the plasticizer, however in P.4, line 141 and 143, o-NPPE is given. In Conclusion section also o-NPPE was listed. The Authors should carefully checked all manuscript.

In Fig.1 the Authors presented experimental results, however they applied spline line to represent the concentration vs. time dependence. Because the Authors used Eq. (1) for description of the transport kinetics, the results of Eq.(1) fitting should be presented in this figure. It should be noted that concentration of the receiving solution vs. time dependence can be also easily calculated from the transformed Eq.(1).

P.5.line 155. How the Authors explain presented supposition that the membrane possess pores?

Reviewer 2 Report

Manuscript NO: 905342

Selective Removal of As(V) Ions from Acid Mine Drainage Using Polymer Inclusion Membranes

In this study, the authors developed a method for selective removal of As(V) using simulated wastewater containing As(V) and sulphate. Finally tested the possible selective removal of As(V) from real AMD. Generally, the work conducted is good but for publication, the authors still need to address the following two important issues:

  1. o-nitrophenyl octyl ether is one of the phthalate ester group which is toxic, the authors said nothing about its leachability during the use. What about the organic composition of the treated AMD water, and need to be addressed. Plasticisers are stable in between pH 5 and 7.

  1. The authors demonstrated the selective removal of As(V) from real AMD, however, nothing was said about the removal of sulphate and other metals from AMD. After removing arsenic it is impossible to discharge the treated AMD which is composed of a high concentration of metals and sulphate to the environment. Authors need to demonstrate possible removal of the pollutants available in AMD and recommend the possible use of the final treated AMD water, based on its composition.

Line 68: please delete “from“ which is repeated

Reviewer 3 Report

1. According to your reference number 28, H2SO4 transport through the membrane could be expected. Have you experienced this coupling phenomena in your study?

  1. Is not clear if you have kept a constant value the pH of the source phase (lines 103 to 105). Have you used a buffer solution?
  2. Could you explain more into detail the transport mechanism? Why the pH gradient stimulated diffusion? Could you comment more on the type of extractant you have used?
  3. You should be consistent in the way you report membrane composition throughout the manuscript. Please stick to wt% values of each component.
  4. What is each aqueous phase composition at the end of the extraction?
  5. Could you explain why the flux through the membrane decreased at higher o-NPOE concentrations than 1.0 cm3 o-NPOE/1.0 g CTA? Are the membranes still flexible/usable? What would happened if you would increase also the carrier concentration at the same time? Do you have SEM or TEM results?

The paper could be improved with discussion on transport mechanism. One open issue is the H2SO4 transport through the membrane that was reported in the literature in similar systems (https://doi.org/10.1021/es030422j)

Round 2

Reviewer 2 Report

  1. See METHOD 8061A: “1.5 Phthalate esters will hydrolyze below pH 5 and above pH 7. The amount of hydrolysis. will increase with increasing or decreasing pH and with longer contact times.” You better do the mass loss test and under similar condition, you removed As(V), which is acidic condition. Not at neutral condition using deionized water. Note that previously used in the PIM system does not mean that phthalates are not leaching, it is well known that phthalates are toxic. Thus, it is not necessary to use for the pilot or industrial scale if these materials are not properly bound to the polymers. The statement you provide in this article will be valuable for future work.

  1. L130-132: Note that at neutral pH phthalates are stable, why did you test the PIM at neutral pH while your experiment for As(V) removal is at acidic condition? You better demonstrate under the similar condition the stability of PIM under the condition you have removed As(V). The Phthalate may be properly bound with the polymer membrane, as the result, no more leached out please check and add into the manuscript your experimental finding that conducted under similar conditions.
  2. No post-remediation statement is available on L299-306 for the other metal removal in the manuscript. 
